# A Comparative Study of Chemical Profiling and Bioactivities between Thai and Foreign Hemp Seed Species (*Cannabis sativa* L.) Plus an In-Silico Investigation

**DOI:** 10.3390/foods13010055

**Published:** 2023-12-22

**Authors:** Suthinee Sangkanu, Thanet Pitakbut, Sathianpong Phoopha, Jiraporn Khanansuk, Kasemsiri Chandarajoti, Sukanya Dej-adisai

**Affiliations:** 1Department of Pharmacognosy and Pharmaceutical Botany, Faculty of Pharmaceutical Sciences, Prince of Songkla University, Hat Yai, Songkhla 90112, Thailand; suthinee.9938@gmail.com (S.S.); jiraporn.kha@psu.ac.th (J.K.); 2Pharmaceutical Biology, Department of Biology, Friedrich-Alexander-Universität Erlangen-Nürnberg (FAU), 91058 Erlangen, Germany; thanet.pitakbut@fau.de; 3Computational Structural Biology Unit, RIKEN-Center for Computational Science, Chuo, Kobe 650-0047, Japan; 4Traditional Thai Medical Research and Innovation Center, Faculty of Traditional Thai Medicine, Prince of Songkla University, Hat Yai, Songkhla 90112, Thailand; sathianpong.p@psu.ac.th; 5Department of Pharmaceutical Chemistry, Faculty of Pharmaceutical Sciences, Prince of Songkla University, Hat Yai, Songkhla 90112, Thailand; kasemsiri.c@psu.ac.th

**Keywords:** *Cannabis sativa*, hemp seed, antibacterial activity, α-glucosidase activity, anti-diabetes, chemical profiling, in silico study

## Abstract

Hemp (*Cannabis sativa* L.) is a plant widely used by humans for textiles, food, and medicine. Thus, this study aimed to characterize the chemical profiling of 12 hemp seed extracts from Thai (HS-TH) and foreign (HS-FS) samples using gas chromatography-mass spectrometry (GC–MS). Their antibacterial activity and α-glucosidase inhibitory activity were assayed. Linoleic acid (17.63–86.53%) was a major component presented in Thai hemp seed extracts, while α,β-gluco-octonic acid lactone (30.39%), clionasterol (13.42–29.07%), and glyceryl-linoleate (15.12%) were detected as the main metabolites found in foreign hemp seed extracts. Furthermore, eight extracts from both Thai and foreign hemp seed exhibited antibacterial activity against *Staphylococcus aureus*, *Staphylococcus epidermidis*, Methicillin-resistant *Staphylococcus aureus*, and *Cutibacterium acnes*, with MIC values ranging from 128 to 2048 µg/mL. Interestingly, the ethanol extract of Thai hemp seed (HS-TH-2-M-E) showed superior α-glucosidase inhibition (IC_50_ value of 33.27 ug/mL) over foreign species. The combination between Thai hemp species (HS-TH-2-M-E) and acarbose showed a synergistic effect against α-glucosidase. Furthermore, the docking investigation revealed that fatty acids had a greater impact on α-glucosidase than fatty acid esters and cannabinoids. The computational simulation predicts a potential allosteric binding pocket of guanosine on glucosidase and is the first description of gluco-octonic acid’s anti-glucosidase activity in silico. The findings concluded that Thai hemp seed could be used as a resource for supplemental drugs or dietary therapy for diabetes mellitus.

## 1. Introduction

*Cannabis sativa* L., or hemp, is a member of the Cannabinaceae family that can grow up to five meters in height during a 4- to 6-month growing period. Hemp has been one of the oldest medicinal plants, cultivated for more than 4500 years for different purposes such as fiber and oil. It is native to Western and Central Asia [1]. Nowadays, more hemp is being grown for industrial, medical, and recreational uses. Iran is one of the major habitats for the cultivation of valuable plants, including hemp; therefore, there is a high degree of genetic variation [2]. The European Union has seen a significant increase in cannabis growing for industrial uses within the last ten years. It is permitted to grow and supply approved *C. sativa* plants from seeds and fiber as long as their Δ^9^-tetrahydrocannabinol (THC) content does not exceed 0.3% [3,4]. Hemp is of interest in the Asia region, and cannabis research is of rising interest in Malaysia. Moreover, there have been reports of criminal cannabis production from other tropical nations, such as India, Indonesia, and Sri Lanka [5]. In 2018, Thailand’s Ministry of Public Health permitted the cultivation of hemp for both industrial and medicinal uses. Hemp can be widely cultivated in Thailand; however, controlling and developing hemp varieties are still very important. Because of the interesting properties of hemp, it is a good strategy to recognize all the opportunities that the global industry and academic community present.

Hemp seed consists of 20–25% protein, 20–30% carbohydrate, 25–35% oil, and 10–15% insoluble fiber [6]. It is used as a medicinal plant because it contains several physiologically active substances, including flavonoids, polyunsaturated fatty acids, terpenoids, and cannabinoids [7]. Additionally, oils contribute significantly to health by serving as a source of oil-soluble vitamins and nutritional value [8]. Hence, several researchers have investigated the various biological activities of this plant. Hemp seed species containing cannabinoids have been reported to have antibacterial activity against Gram-positive bacteria (*Staphylococcus aureus*, *Enterococcus faecalis*, and *Streptococcus pneumoniae*) and Gram-negative bacteria (*Salmonella enterica* and *Pseudomonas aeruginosa)* [9,10]. Moreover, oil derived from the hemp’s seed has shown some antibacterial properties because of myrcene, α-pinene, and limonene [11]. The investigation conducted by Mikulcová et al. [12] led to the suggestion that the increased antibacterial activity of hemp seed oils could be attributed to their higher α-linolenic acid (ALA) content.

According to the International Diabetes Federation, diabetes mellitus affects approximately 387 million people worldwide and caused 4.9 million fatalities in 2014. As a result, it is becoming a major public health concern with enormous societal and health-care costs [13]. Ninety percent of diabetes cases are type 2 diabetes mellitus (T2DM), which is typified by people with postprandial hyperglycemia linked to either poor insulin secretion, resistance to insulin, or both. The method of managing type 2 diabetes is the use of digestive enzyme inhibitors, such as the intestine’s α-glucosidase, which catalyzes the breakdown of complex carbs into easily digestible monosaccharides [14]. Numerous bioactive substances found in hemp seed, such as polyphenols, alkaloids, cannabinoids, and terpenoids, have been shown to have protective effects against a number of illnesses, such as hypertension, diabetes, cancer, and oxidative stress [15]. The study by Zengin et al. [16] demonstrated that the essential oils from the *Cannabis* aerial part exhibited antidiabetic activities against the α-glucosidase enzyme with a value of 3.77 mmol ACAE/g oil; however, this essential oil did not inhibit α-amylase. Presently, a computational method called “molecular docking” attempts to forecast possible interactions between a protein and one or more ligands. It is highly helpful in directing the choice of bioactive compounds for in vitro experiments. Numerous molecular docking studies on the various kinds of metabolites found in the cannabis plant—cannabinoids; terpenes; polyphenols; flavonoids; lignanamides; alkaloids; vitamins; and proteins—have been tested against several enzymes; but α-glucosidase was not present [17].

Therefore, this research aims to study the chemical composition of *Cannabis sativa* from Thai and foreign species. It also aims to investigate the biological activities, including antibacterial and α-glucosidase inhibition, of hemp seed extracts alone and in combination with acarbose. Finally, this study also includes molecular docking studies of secondary metabolites that were identified in different species of *Cannabis sativa* L. on the α-glucosidase targets.

## 2. Materials and Methods

### 2.1. Plant Material

Four samples of *Cannabis sativa* were subjected to extraction in this study. Two samples of Thai hemp seeds: RPF1 (HS-TH-1) was courtesy of T-Basa Company Limited, Raman District, Yala Province, Thailand and RPF3 (HS-TH-2) was courtesy of the Highland Research and Development Institute, Mueang Chiang Mai District, Chiang Mai, Thailand (A Public Organization). Two samples of foreign hemp seed, FO#2 (HS-FS-1) and FO#3 (HS-FS-2), were purchased from Sri Trang Agro-Industry Public Company Limited Sri Trang Rubber & Plantation Company Limited, Mueang Chiang Mai District, Chiang Mai, Thailand and courtesy of T-Basa Company Limited, respectively. Prior to processing, seeds were cleaned with water and dried. The dried samples were kept in a sealed container until extraction.

### 2.2. Hemp Seed Extraction

About two kilograms of each sample were extracted using an oil-pressed extractor, and the temperature of the oil extractor was controlled not to exceed 40 °C. Hemp seed oil (HS-O) was obtained in this step and labeled as HS-TH-1-O, HS-TH-2-O, HS-FS-1-O, and HS-FS-2-O. Marc is divided into 2 parts for 80% ethanol and hexane extraction. Materials were macerated with 80% ethanol and hexane at room temperature for 3 days and repeated 3 times. The extract was filtered, and the solvent was evaporated using a rotary evaporator to obtain four samples of 80% ethanol crude extract (HS-TH-1-M-E, HS-TH-2-M-E, HS-FS-1-M-E, and HS-FS-2-M-E) and four samples of hexane crude extract (HS-TH-1-M-H, HS-TH-2-M-H, HS-FS-1-M-H, and HS-FS-2-M-H). The extracts were stored at −20 °C until use.

### 2.3. Gas Chromatography-Mass Spectrometry (GC–MS) Analysis

GC–MS analyses of hemp seed oil and extract were performed using an Agilent Technologies 7890 B (GC) equipped with a 5977B Mass Selective Detector (MS). Separation was achieved on a VF WAXms capillary column (30 m × 250 × 0.25 μm) coupled with helium gas as the carrier at a flow rate of 1 mL min^−1^. The column temperature was programmed initially at 60 °C, which was increased to 160 °C at 10 °C min^−1^ and further increased to 250 °C at 2.5 °C min^−1^, holding time for 15 min. The mass spectrometer was operated in the electron ionization mode at 70 eV with a source temperature of 230 °C, with continuous scanning from 35 to 500 m/z. The chemical constituents were identified by comparing their mass spectral data with those from the Wiley library and NIST14 libraries.

### 2.4. Antibacterial Activity

#### 2.4.1. Test Microorganisms

The following test microorganisms were used in this study: *Staphylococcus aureus* (ATCC 25923), *Staphylococcus epidermidis* (TISTR 517), *Cutibacterium* (*Propionibacterium*) *acnes* (DMST 14916), Methicillin-resistant *Staphylococcus aureus*, MRSA (DMST 20654), *Pseudomonas aeruginosa* (ATCC 27853), and *Escherichia coli* (ATCC 25922). Bacterial strains were cultured on Mueller-Hinton agar (MHA) and incubated at 37 °C for 18–24 h, except *C. acnes* (DMST 14916), which was cultured on Brain Heart Infusion Agar (BHIA) in an anaerobic jar at 37 °C for 96 h. The suspensions of bacteria were prepared using Mueller-Hinton broth (MHB) or Brain Heart Infusion broth (BHIB), then adjusted to equal the turbidity of 0.08–0.1 at OD_625_ (10^8^ CFU/mL) with sterile normal saline solution (NSS). For the broth dilution test, the bacterial suspensions were further diluted 1:100 using MHB or BHIB to obtain approximately 10^6^ CFU/mL.

#### 2.4.2. Preliminary Antibacterial Test of Hemp Seed Oil and Extract

Hemp seed oils and extracts at a final concentration of 2 mg/mL were preliminary tested against tested bacteria by a colorimetric broth microdilution method in 96-well microtiter plates [18] with some modifications. Briefly, oils and extract stock solutions (200 mg/mL) were diluted to 4 mg/mL with MHB or BHIB. Then, 50 μL of each sample was transferred to each well in triplicate. Fifty microliters of tested bacterial inoculum (10^6^ CFU/mL) were added, giving a final extract concentration of 2 mg/mL. Plates of all bacterial strains were incubated at 37 °C for 15 h, except *C. acnes*, which was incubated in an anaerobic jar at 37 °C for 96 h. Then, 20 μL of resazurin solution (0.1%) was added into each well and further incubated for 3 h as a modified from [19]. The negative result was the color change of resazurin from purple to pink. A blue or purple color indicated inhibition of bacterial growth, considered a positive result. Oxacillin and norfloxacin (10 μg/mL) were used as positive controls for Gram-positive and Gram-negative bacteria, respectively.

#### 2.4.3. Determination of Minimum Inhibitory Concentration (MIC) and Minimum Bactericidal Concentration (MBC)

The CLSI [18,19], with some modifications, was used to evaluate hemp seed oils and extracts for their MICs and MBCs. This study was carried out to assess the MICs and MBCs of the extracts that showed antibacterial activity at 2 mg/mL. Ten concentrations of samples were 2-serially diluted, ranging from 2048 to 4 µg/mL, and they were analyzed in triplicate. The MIC endpoint was the lowest concentration of each extract at which there was no color change (blue color) of resazurin. The MBC was determined by sub-culturing all positive wells onto MHA or BHIA plates and incubating under appropriate conditions. The MBC endpoints were defined as the lowest concentration of extract that showed no visible growth. All tests were performed in triplicate.

### 2.5. α-Glucosidase Inhibition Activity

Hemp seed oils and extracts were investigated for their α-glucosidase inhibitory activity [20]. A 96-well microtiter plater was filled with 0.01 M phosphate buffer pH 7.0 (50 µL), samples or standard solution (50 µL), and α-glucosidase enzyme solution (50 µL). The mixture solution was incubated at 37 °C for 2 min. Then, the substrate solution, *p*-nitrophenyl-α-D-glucopyranoside solution (50 µL), was added. The end product (yellow color) of *p*-nitrophenol (*p*NP) was used to assess the enzymatic inhibition by visible light at 405 nm of VARIOSKAN LUX (Thermo Scientific, Waltham, MA, USA). Equation (1) was used to convert the product absorption to velocity (V). Equation (2) was then used to calculate the percentage of inhibition using the velocity. Additionally, the IC_50_ value, which was produced from the calibration curve of percentages of inhibition at 5 different sample concentrations, was used to measure the capability of the sample.
Velocity = ∆ Absorbance at 405 nm/∆ Time(1)
% Inhibition = [(V_control_ − V_sample_)/V_control_] × 100(2)

### 2.6. Combination Study

Using enzyme inhibitors or combining inhibitors with different drug strategies to overcome drug resistance and decrease toxicity. In this study, the combination effect of hemp seed extracts with acarbose was evaluated. The IC_50_ values of hemp seed extracts and acarbose against α-glucosidase were calculated based on dose-response curves. Then, a series of 5 concentrations of hemp seed extracts (at 4IC_50_, 2IC_50_, IC_50_, 0.5IC_50_, and 0.25IC_50_) were prepared and combined with a series of 5 concentrations of acarbose in equal volumes to be used to examine the combined inhibitory effects of hemp seed extract and acarbose on α-glucosidase. Data were analyzed using CompuSyn software (CompuSyn Version 1.0; https://www.combosyn.com/) (accessed on 30 October 2023) to determine the combination index (CI) based on the median-effect principle developed by Chou and Talalay. The CI values of drug-drug combinations were characterized as synergistic (CI < 1), additive (CI = 1), or antagonistic (CI > 1) [21].

### 2.7. Molecular Docking Simulation

#### 2.7.1. Docking Preparation

For ligands, all chemical structures were downloaded from the PubChem database (https://pubchem.ncbi.nlm.nih.gov, accessed on 23 September 2023). The 3D structures of all compounds were obtained through geometry and energy forcefield (mmff94) using the Open Babel program version 3.1.0 [22]. Finally, all files were converted to pdbqt format for docking simulation. On the other hand, for protein structure, glucosidase enzyme crystal structure (PDB ID: 3a4a) were obtained from the RCSB PDB database (https://www.rcsb.org, accessed on 23 September 2023) [23]. Autodock Tools (version 1.5.6) was used to prepare the protein structure for the docking simulation [24].

#### 2.7.2. Docking Experiment

Following our previous reports [25,26], Autodock Vina (version 1.2.5) was used to perform a docking experiment [27]. The glucosidase catalytic domain was set as a docking site or grid box. The active site was determined and presented as a three-dimensional grid with X = 21.2, Y = −7.5, and Z = 24.3, respectively, and the size was set at 18 Å × 18 Å × 18 Å. Nearly all docking parameters were set as default values except exhaustiveness and number of models. The exhaustiveness and number of models were adjusted to 10 and 20 accordingly. However, the grid box size was extended to 70 Å × 90 Å × 90 Å, covering the entire glucosidase when the docking protocol was to dock guanosine, aiming to discover an unknown allosteric binding pocket of guanosine. All docking protocols used in this study were validated via a re-docking approach. Only a docking protocol that provided less than 3 Å RMSD value after re-docking the extracted native ligand back into its original position (glucosidase’s active site) was used [25,26].

As described earlier, each docking experiment generated twenty docking poses. However, only one pose was selected. The selection criterion was based on the scientific assumption that a similar compound has a similar binding position. In the case of guanosine, like the grid gride box extension, the docking pose selection criterion was adjusted. Since the aim is to discover a guanosine allosteric binding site, the authors selected one pocket with the most docking poses as a promising guanosine binding site on glucosidase.

#### 2.7.3. Post-Docking Analysis, Modeling, and Statistical Analysis

After selecting docking poses, single-point energy estimation via Autodock 4 (version 4.2.6) was used to rescore the selected docking poses for further energetical analysis [24]. LibreOffice, version 7.5.2.2, was utilized to calculate, compare, visualize, and establish a linear model. For the non-linear model, MyCurveFit, an online tool (https://mycurvefit.com, accessed on 10 October 2023), was used to identify the best-fit model. Finally, 2D and 3D interaction diagrams between selected poses and amino acids in the binding domain were generated using Liplot-plus (version 2.2.8) [28] and Chimera (version 1.17.3) programs [29].

## 3. Results

### 3.1. Hemp Seed Extraction

During the screw-pressing extraction process, samples of extracted oils were collected. Figure 1 shows the color of the collected oils (HS-TH-1-O, HS-FS-1-O, and HS-FS-2-O) was almost yellow, except HS-TH-2-O received a green color. Marc obtained it after oil extraction, which used 80% ethanol and hexane maceration. The color of 80% ethanolic extracts varied from brown (HS-TH-1-M-E and HS-TH-2-M-E), green (HS-FS-1-M-E, HS-TH-2-M-H, and HS-FS-2-M-H), to yellow (HS-FS-2-M-E), while some hexane extracts presented colorless (HS-TH-1-M-H and HS-FS-1-M-H).

The yield of oils and extracts from hemp seed is shown in Appendix A. It was found that the foreign hemp seed HS-FS-2-O produced the highest yield of oil (19.16% *w*/*w*), followed by HS-FS-1-O (15.26% *w*/*w*), HS-TH-2-O (15.21% *w*/*w*), and HS-TH-1-O (14.85% *w*/*w*). The marc was extracted with 80% ethanol and hexane, yielding less than 10% in all samples.

### 3.2. GC–MS Analysis

The chemical composition of hemp seed oils and extracts was determined by GC–MS techniques, and the results are shown in Table 1. The main component (asterisk symbol) of Thai hemp seed extracts (HS-TH-1-O, HS-TH-1-M-H, HS-TH-1-M-E, HS-TH-2-O, HS-TH-2-M-H, and HS-TH-2-M-E) was linoleic acid, which together comprised 20.09, 22.93, 35.13, 86.53, 66.24, and 34.08% of the total compounds. In addition, a high amount of linoleic acid is also present in foreign hemp seed extracts (HS-FS-1-M-H, 17.63%). Hemp seed oils and extracts contained amounts of palmitic acid (0.28–10.21%), palmitic acid ethyl ester (1.18–8.57%), linoleic acid ethyl ester (1.08–20.61%), and stearic acid ethyl ester (2.37–3.92%). In addition, some fatty acids such as α-linolenic acid, oleic acid, and stearic acid were found in HS-FS-2-M-E (1.52%), HS-FS-1-O (3.51%), and HS-FS-1-M-H (1.73%), respectively.

The ethanolic extracts of foreign hemp seed, HS-FS-1-M-E and HS-FS-2-M-E, had 1-linoleoyl glycerol (15.12%) and α,β-gluco-octonic acid lactone (30.39%) with a high composition, respectively. Phytosterol, namely clionasterol, is represented by the major compounds HS-FS-1-O (29.07%), HS-FS-2-O (22.32%), and HS-FS-2-M-H (13.42%). It was found in other extracts in the range of 1.44–6.66%. Other interesting phytosterols in hemp seed extracts were campesterol (1.04–4.81%), (3methyl,24R)-ergost-5-en-3-ol (3.78–7.61%), lanosterol (1.91–5.22%), stigmasterol (1.41–1.45%), and (3b,24Z)-Stigmasta-5,24(28)-dien-3-ol (2.08–5.95%), γ-sitostenone (1.13–2.81%), and cycloartenol (1.74–3.48%).

Cannabinoids are naturally occurring compounds found in hemp seed oils. In this study, we found that HS-TH-1-O composed ∆^9^-tetrahydrocannabivarin (THCV) (1.20%), cannabidiol (CBD) (0.35%), ∆^9^-THC (dronabinol) (3.36%), and cannabinol (CBN) (1.32%). HS-TH-1-M-H found only CBN (0.51%), while HS-TH-1-M-E, HS-TH-2-O, HS-TH-2-M-H, and HS-TH-2-M-E did not contain cannabinoids. Two samples of foreign hemp seed extracts showed different cannabinoid profiles. We found only CBD (13.39%) in the oil of the FO#2 (HS-FS-1-O) sample, while THCV, CBD, and ∆^9^THC were produced by the FO#3 sample. The oil part of FO#3 (HS-FS-2-O) contained THCV (5.62%), CBD (0.53%), and ∆^9^THC (18.20%). HS-FS-2-M-H and HS-FS-2-M-E showed similar cannabinoids in that they presented THCV (1.05, 2.73%) and ∆^9^THC (2.70, 3.81%).

### 3.3. Preliminary Antibacterial Activity

The antibacterial activity of hemp seed oils and extracts against six bacterial strains was tested. The results of the preliminary test at a concentration of 2 mg/mL are shown in Appendix A. Oil and ethanolic extracts (HS-TH-2-O and HS-TH-2-M-E) of RPF-3 inhibited the growth of all Gram-positive bacteria, namely *S. aureus*, *S. epidermidis*, MRSA, and *C. acnes*. The hexane extract, HS-TH-2-M-H, also inhibited Gram-positive bacteria except *S. epidermidis*. Ethanolic extract (HS-TH-1-M-E) of RPF-1 had antibacterial activity against all Gram-positive bacteria, but oil (HS-TH-1-O) and hexane extract (HS-TH-1-M-H) inhibited only *S. aureus*. For foreign hemp seed samples, ethanolic extracts (HS-FS-1-M-E and HS-FS-2-M-E) were effective against *S. aureus* and *C. acnes* but had no antibacterial activity from oil (HS-FS-1-O and HS-FS-2-O) or hexane extracts (HS-FS-1-M-H and HS-FS-2-M-H).

### 3.4. Determination of Minimal Inhibitory Concentration (MIC) and Minimal Bactericidal Concentration (MBC)

The minimal inhibitory concentration (MIC) and minimal bactericidal concentration (MBC) of the active oils and extracts were determined in the concentration range of 2048–4 µg/mL. The MIC values were obtained in the range of 2048–128 µg/mL and the MBC values in the range of 256 to >2048 µg/mL (Table 2). *S. aureus* and *S. epidermidis* were the most susceptible bacteria in this study. Hemp seed oil from a Thai sample, RPF-3 (HS-TH-2-O), possessed the most effective agent against *S. aureus* with a MIC value of 128 µg/mL. In addition, HS-TH-2-O also inhibited *S. epidermidis*, MRSA, and *C. acnes* at a concentration of 256 µg/mL. For *S. epidermidis*, the ethanolic extract from the Thai hemp seed sample, RPF-1 (HS-TH-1-M-E), showed the most inhibition at a MIC value of 128 µg/mL. This extract also inhibited *S. aureus*, MRSA, and *C. acnes* at 512 µg/mL. Only two extracts from foreign hemp seed samples (HS-FS-1-M-E and HS-FS-2-M-E) showed antibacterial activity against *S. epidermidis* (MIC values of 256 and 2048 µg/mL) and *C. acnes* (MIC values of 2048 and 1024 µg/mL).

### 3.5. Screening and IC_50_ on α-Glucosidase Inhibition of Hemp Seed Oils and Extracts

The screening of α-glucosidase inhibition results is shown in Table 3, and the ethanolic extracts of all samples (HS-TH-1-M-E, HS-TH-2-M-E, HS-FS-1-M-E, and HS-FS-2-M-E) exhibited strong inhibitory activity at 2 mg/mL. In addition, they showed significant inhibition at a concentration of 0.25 mg/mL. The strong active extracts were brought to further IC_50_ determination, and we found that all extracts displayed strong inhibitory activity over the standard drug, acarbose (IC_50_ = 285.92 µg/mL). The most active extract was HS-TH-2-M-E (IC_50_ = 33.27 µg/mL), followed by HS-TH-1-M-E (IC_50_ = 42.72 µg/mL), HS-FS-2-M-E (IC_50_ = 67.44 µg/mL), and HS-FS-1-M-E (IC_50_ = 94.09 µg/mL) (Table 3).

### 3.6. Combination Study on α-Glucosidase Inhibitory of Hemp Seed Extract and Acarbose

Combinations of hemp seed extract and acarbose were expected to reduce the side effects of acarbose and showed higher α-glucosidase inhibition activity compared to a single extract or drug. As suggested in our findings above, HS-TH-2-M-E shows the most potent extracts against α-glucosidase, so it was chosen for a combination study. When acarbose at different concentrations (0.25–4 IC_50_) was combined with HS-TH-2-M-E at high concentrations (2–4 IC_50_), CI values in the range of 0.80–0.99 were obtained. In addition, when combined with acarbose (2–4 IC_50_) with HS-TH-2-M-E at low concentrations (0.5–1 IC_50_), CI values were calculated in the range of 0.70–0.99. These CI values indicated synergistic effects on α-glucosidase inhibition activity between acarbose and HS-TH-2-M-E (Table 4).

### 3.7. Molecular Docking and a Mathematical Model

This work used molecular docking for two purposes. The first and main purpose is to use an in silico experiment to support and explain an experimental α-glucosidase inhibition activity. We examined the relationship between the obtained α-glucosidase inhibition experimental data and detected metabolites from the GC–MS phytochemical profile in each extract. In total of 15 compounds, there were five free fatty acids (palmitic acid, stearic acid, oleic acid, linoleic acid, and linolenic acid), five fatty acid esters (ethyl-palmitate, ethyl-oleate, ethyl linoleate, glyceryl-linoleate, and ethyl-linoleate), two cannabinoids (THC and THCV), one phenolic compound (DTBP), and two glycosides (guanosine and gluco-octonic acid lactone). Each docked compound generated twenty poses. However, only one pose was selected as the most promising. There were two selection criteria. The first was based on the previous mode of action reports, and the second was a hypothesis that components that share essential chemical features have a similar binding conformation. After selecting the best docking conformation, we evaluated the energy pattern of the selected docking poses via re-scoring function analysis. The obtained energy function were used to compare the current study simulation to the previous experimental data [10], determining the reliability of the simulation. This step only applied to free fatty acids and their ester since there were insufficient experimental data for other metabolites. The secondary aim of the simulation is to predict the theoretical possible binding site of new and previously known glucosidase inhibitors detected by a GC–MS analysis.

#### 3.7.1. Molecular Docking and Mathematical Model of Free Fatty Acids and Esters

First, we validated our docking protocol via re-docking validation and received an RMSD value of less than 3 Å (indicating a reliable protocol). More information on the validation can be found in a Appendix A. As presented in Figure 2A, molecular docking showed that all fatty acids and their esters inserted a polar head group (either carboxylic or carboxyl groups) into a catalytic pocket, and the position of polar heads was in proximity to a glucose molecule, a native ligand from a crystal structure. Noticeably, molecular docking also revealed that a fatty acid could insert a polar head group deeper inside the catalytic pocket than its ester counterparts. For example, Figure 2C–E showed a structural alignment comparing linoleic acid, ethyl-linoleate, and glyceryl-linoleate. Even though a carboxyl group of glyceryl-linoleate was not as shallow as linoleic acid, hydroxy groups from a glyceride moiety were. In this study, the authors only showed linoleic derivatives as an example because they are the main component found in most extracts based on GC–MS analysis. However, a complete structural alignment of all fatty acids found in this study was provided in a Appendix A.

Figure 3A demonstrates a re-scoring docking function in each category, starting from the final intermolecular energy (1), the final total internal energy (2), the torsional free energy (3), the unbound system’s energy (4), and the estimated free energy of binding (5) of the summarized re-scoring function of five fatty acids and their esters. Only category 3, torsional free energy, agreed with the previous experimental report focusing on the different activities between fatty acids and their ester counterparts. Even the observed difference between free and ester fatty acids was not statistically significant. The summarized torsional free energy of all fatty acid esters exhibited a higher value than free fatty acid, indicating less stability (weaker interaction and inhibitory activity). This matched a previous report [30]. Then, we further evaluated the torsional free energy of each fatty acid according to carbon length and degree of unsaturation, as presented in Figure 3B. The torsional free energy of all fatty acids showed a similar trend: all ester forms had a higher energetic value than the free acid form.

Later, we evaluated a relationship between all re-scoring docking functions (1 to 5, as mentioned in the previous paragraph) of all fatty acids found in the current study and previous experimental data based on the aliphatic length and degree of unsaturation [30]. Interestingly, only the final total internal energy correlated to the previous experimental data, while the others did not. The detailed evaluation was provided in the Appendix A.

After that, the authors plotted the final total energy and IC_50_ values (from the previous experimental report) against the fatty acid length and unsaturation degree, as shown in Figure 3C,D. The authors obtained a sufficient correlation (R^2^ value of 0.6828) from the final internal energy and fatty acid relationship and a good correlation (R^2^ value of 0.8797) between IC_50_ values and fatty acids. As Figure 3C,D demonstrated, both figures exhibited a similar pattern (decay exponential function), showing that an additional aliphatic length and degree of unsaturation contributed to a lower final total energy and lower IC_50_ values. This finding indicated that the re-score docking energy and the anti-glucosidase experimental data for fatty acids behave similarly. However, the re-scoring docking function of the oleic acid (C18:1) individually did not agree with the decay exponential pattern, while the other fatty acids went along. Therefore, the authors excluded a re-score function of the oleic acid’s total final energy and re-evaluated the correlation. As shown in Figure 3C, after exclusion, the R^2^ value increased to 0.9784. This information indicated that the docking program could not provide the correlated total internal energy to previous experimental data for oleic acid.

Therefore, we utilized the decay exponential model to statistically examine the relationship between the final internal energy and actual experimental data from a previous report. The model and experimental data revealed a sufficient linear relationship with an R^2^ value of 0.8582 (Figure 3E). Therefore, this indicated that the estimated final total internal energy could be used to predict the trend of an experimental outcome for fatty acid derivatives. To the authors’ best knowledge, this is the first study to demonstrate a relationship between a modeled re-scoring docking function and actual experimental data for a series of free fatty acids.

#### 3.7.2. Molecular Docking of THC and THCV

A similar procedure as mentioned above was applied here to THC and THCV. Figure 4A shows the selected docking poses of both compounds. THC and THV had a nearly identical structural alignment and shared similar interaction residues in an α-glucosidase active site, as shown in Figure 4A,B. These two cannabinoids could dock at an active site of α-glucosidase. However, their poses were shallower than a glucose molecule (native ligand), as shown in Figure 4A.

Later, we rescored a docking energetical function to predict the experimental anti-glucosidase activity of THCV compared to THC. Table 5 exhibits a detailed re-scoring function of both THC and THCV. The authors applied prior knowledge found in the fatty acid case since cannabinoids share part of the biosynthetic pathway with fatty acids. Based on this table, the authors focused on two energetical values contributing to the difference between THCV and THC. The first is torsional free energy, and the second is final total internal energy. Both energies exhibited a vital role in fatty acid and its ester analysis earlier and were related to previously reported experimental data [30]. However, both values were in contradiction. In summary, THCV exhibited lower torsional free energy than THC but had a higher final total internal energy. Even the estimated free binding energy and inhibition constant (Ki), as a summarized value of all energetical functions, agreed with the torsional free energy, indicating better activity from THCV. However, these two functions did not correlate with experimental data, as shown earlier in the fatty acid case. Based on their chemical structure, the only difference between these cannabinoids is the aliphatic chain length. These chemical variants did not contribute to a conventional intermolecular bond like a hydrogen bond. Only non-bond close contacts were observed based on the 2D interaction diagram (Figure 4B,C). Therefore, the authors used the current study’s findings on fatty acids as a guideline. Based on the current information, the authors predicted that THCV would likely have weaker activity than THC due to a higher total internal energy from a re-coring function analysis.

#### 3.7.3. Molecular Docking of DTBP

Noticeably, DTBP was uniquely found in the HS-TH-2-M-E as a minor component, as presented earlier in a GC–MS analysis. The best pose (the lowest binding energy) suggested by Autodock Vina showed that DTBP forms a hydrogen bond with ARG442 in the glucosidase catalytic domain and seven non-bond contacts with seven amino acids, including TYR158, PHE159, PHE303, ARG315, ASP352, GLN353, and GLU411 (Figure 5).

#### 3.7.4. Molecular Docking of Guanosine: Proposing a Possible Allosteric Binding Site

In 2020, Ogasawara et al. reported that guanosine exhibited non-competitive inhibitory behavior against amylase in an in vitro enzyme kinetic assay [31]. This information from Ogasawara and the team indicated that guanosine is bound to an enzyme in a non-catalytic pocket, an allosteric site. From the literature, it is possible that an amylase non-competitive inhibitor can also inhibit glucosidase in the same manner [32]. Therefore, it is likely that guanosine might inhibit the glucosidase similarly, and this is the authors’ hypothesis. To prove the hypothesis, the authors modified the authors’ docking protocol to cover the entire glucosidase enzyme and perform molecular docking. Since the docking protocol was modified, the authors re-validated the new protocol. This particular docking aims to identify the binding pocket of the ligand of interest, not a ligand conformation like conventional docking. Therefore, the evaluation criterion shifted from an RMDS value of less than 3 Å to the highest docking poses in the same pocket. The modified docking protocol could re-dock a glucose molecule (a native ligand) back into an active site (the original pocket) with the highest docking poses. It indicated that the entire enzyme structure could be docked to predict a ligand-binding site. The authors provided information regarding the entire enzyme structure docking validation in a Appendix A.

Figure 6 demonstrates six possible allosteric bindings of guanosine interacting with glucosidase, as proposed by Autodock Vina. The black circle highlighted the top three sites and named clusters 1, 2, and 3 accordingly. Cluster 1 consisted of twelve docking poses (the highest), while clusters 2 and 3 showed only three and two docking poses. Therefore, based on the highest number of docking poses, we proposed that cluster 1 is most likely an allosteric binding pocket of guanosine on α-glucosidase. Then, one of the many docking poses with the lowest docking score of each cluster was selected to examine a ligand-amino acid interaction. Figure 6B,C exhibited representative ligand-amino acid interactions in the binding pocket of each cluster. Notably, molecular docking simulations indicated multiple hydrogen bonds between guanosine and binding residues, suggesting a firm binding.

#### 3.7.5. Molecular Docking of Gluco-Octonic Acid Lactone, Proposing a Competitive Behavior

Different from previous molecular docking, this simulation did not have any experimental data to support it since there were no information on gluco-octonic acid lactone and glucosidase available. Up to date, gluco-octonic acid lactone’s bioactivity information is minimal. Fortunately, gluco-octonic acid lactone is an assembly of glucofuranose (Figure 7A). Therefore, the authors hypothesize that gluco-octonic acid lactone could dock into an active site of α-glucosidase like a glucose molecule. After performing a molecular docking simulation, the authors found that gluco-octonic acid lactone could insert its structure deep inside the glucosidase catalytic domain, like a native molecule (glucose), as shown in Figure 7B. The 2D interaction diagram (Figure 7C) identified four hydrogen bonds with ASP69, GLU277, GLN279, ASP352, and ARG442, and thirteen non-bonding contacts with ASP69, TYR72, HIS112, TYR158, PHE159, PHE178, ASP215, VAL216, GLU277, GLN279, PHE303, ASP352, and ARG442. Three out of four hydrogen bonds were involved with the catalytic residual, except GLN279. Therefore, based on our theoretical simulation, we proposed a competitive interaction between glucosidase and its substrate.

## 4. Discussion

Hemp is an industrial crop that has been extensively studied because of its many uses in the food, medicinal, nutritional, cosmetic, textile, and material industries [15]. Typically, hemp seeds contain more than 30% lipids and approximately 25% protein. They also contain high amounts of dietary fiber, vitamins, and minerals. Hemp seed oil contains more than 80% unsaturated fatty acids and is a rich source of two important polyunsaturated fatty acids: linoleic acid (18:2 omega-6) and α-linolenic acid (18:3 omega-3). The ratio that is accepted as appropriate for consumption is omega 6: omega 3 equal to 3:1 [33]. These unsaturated fatty acids have been linked to anti-inflammatory conditions, obesity, diabetes mellitus, and protection against cardiovascular illnesses [34]. The green color of hemp seed oil extracts comes from the most important pigments, especially chlorophyll a and b. Furthermore, carotenoids are also present in hemp oil. However, the ratio of the carotenoid/chlorophyll proportion varied significantly, depending on the hemp seed variety [35]. The chemical profiles of 12 extracts from 4 hemp seed samples showed the differences between Thai and foreign samples in the comparative study. Most Thai hemp seed extracts had the fatty acid linoleic acid (omega-6) as a major component (20.09–86.53%). However, α-linolenic acid was not detected in all extracts. Foreign hemp seed extracts provided a variety of major compounds, such as linoleic acid, clionasterol, glyceryl-linoleate, and α,β-gluco-octonic acid lactone. Numerous studies have examined and concluded that genotype impacts the chemical composition of hemp seed, particularly fatty acids. According to Irakli et al. [36], the genotype variation less impacts linoleic acid, but α-linolenic acid and oleic acid are more affected by it. However, Galasso et al. [37] demonstrated that the genotype of hemp seed impacted both linoleic acid and α-linolenic acid variability. In addition to the factors mentioned above, the location and climate also impact the components of fatty acids. Taaifi et al. [38] assessed the hemp seed oil composition of two hemp cultivar varieties grown in four distinct Moroccan regions. They found that the observed variability in the composition was related to the growing area, the interaction of climatic conditions, and the cultivar variety. The predominant group of components in foreign hemp seed extracts was phytosterol. Clionasterol was a major component of the extracts HS-FS-1-O, HS-FS-2-O, and HS-FS-2-M-H. Other types of sterols, such as campesterol, cycloartenol, lanosterol, and stigmasterol acetate, were also found. Clionasterol is the most common sterol group found in as much as 76% of hemp seed oil [39]. This study is consistent with the reports of Liang et al. [40] and Montserrat-DeLaPaz et al. [41], who found clionasterol (270–380 mg/100 g of oil) as a main compound in hemp seed oil. Furthermore, Glyceryl-linoleate, or (Z,Z)-9,12-octadeca-dienoic acid, 2,3-dihydroxypropyl ester, was found as a main component in HS-FS-1-M-E. The monoester of glycerin and linoleic acid is used in cosmetics mostly as a skin conditioning agent, emollient, and surfactant emulsifying agent [42]. α,β-Gluco-octonic acid lactone was discovered to be the main component in extracts from foreign hemp seeds (HS-FS-2-M-E).

Hemp extracts have become a powerful alternative in several fields because they show a wide spectrum of activities for bioactive compounds such as antioxidants, anti-inflammatory, neuroprotective, prebiotic, antimicrobial, and antifungal. In this study, we investigated the antibacterial and α-glucosidase inhibitory activities of hemp seed extract. Six extracts from Thai samples and two extracts from foreign samples exhibited antimicrobial activities against Gram-positive bacteria. In a previous study, hemp seed oil extracted using a hexane solvent showed effective growth inhibition of *C. acnes* [43]. When focusing on the main components of the extract, it was found to contain polyunsaturated fatty acids, including linoleic acid, oleic acid, cis-11-eicosenoic acid, palmitic acid, γ-linolenic acid, arachidic acid, palmitoleic acid, and heneicosanoic acid. The fatty acids can act as anionic surfactants and have antibacterial and antifungal properties at low pH [44]. Gram-positive bacteria were more susceptible to hemp seed extracts than Gram-negative bacteria. Gram-negative bacteria typically have outer membranes that act as barriers, making them more resistant to other substances [45].

Inhibition of α-glucosidase is one of the therapeutic approaches for T2DM management, resulting in a carbohydrate digestion delay and reducing glucose absorption. Several studies have reported the effects of herbal medicine on suppressing α-glucosidases [46,47]. According to our present study, hemp seed extracts significantly reduce the activity of α-glucosidase compared to acarbose. Ethanolic extracts from all hemp seed samples exhibited potential activity. Previous research demonstrated that there is a significant relationship between the fatty acid content of the extract and its ability to inhibit α-glucosidase [48]. This study found that HS-TH-2-M-E exhibited the best α-glucosidase inhibitory effect, although other ethanolic extracts contained fatty acids. It may be due to minor components in the extract that supported α-glucosidase inhibitory activity, such as 4-Di-tert-butylphenol (2,4-DTPB). 2,4-DTPB is found in bamboo shoot extract, and it has been reported to act as an anti-diabetic agent that inhibits the activity of the enzymes α-glucosidase and α-amylase [49]. Moreover, when combined with acarbose, HS-TH-2-M-E showed an increased inhibition of α-glucosidase.

In this study, we performed a computational molecular simulation and constructed a mathematical model that was correlated to previous experimental data. Before performing the molecular docking simulation of fatty acids, we varied the setup docking parameters to ensure the accuracy of the computational experiment. In the re-docking validation test, we received an RMSD value of less than 3 Å from the redocking validation test, passing an accepted range [13]. From the model, a free fatty acid with a longer aliphatic chain and a higher degree of unsaturation significantly reduced the final total internal energy during enzyme interaction molecularly—minimal energy led to a more favorable ligand binding and a small observed IC_50_ value. Therefore, in a current study, the model rationalized how hemp seed extracts exhibited different α-glucosidase inhibitions based on metabolomic GC–MS profiles. As shown earlier, linoleic acid (LA) and palmitic acid (PA) are the most abundant fatty acids detected in both RPF extracts, with nearly the same concentration range. However, based on IC_50_ values in this study, we found that HS-TH-2-M-E exhibited stronger inhibitory activity against α-glucosidase than HS-TH-1-M-E. The only difference among these extracts is 2,4-DTBP. Therefore, the information hints that 2,4-DTBP might be a synergistic agent against α-glucosidase. 2,4-DTBP was uniquely found in HS-TH-2-M-E as a minor component. Recently, Sansenya et al. [49] reported that DTBP exhibited a mixed-type inhibitory behavior (predominately non-competitive) against α-glucosidase activity. In Sansenya’s report, a molecular docking experiment was conducted. However, no hydrogen bond was observed between DTBP and α-glucosidase in Sansenya’s study. Even though 2,4-DTBP showed predominantly non-competitive behavior, it also acted as a competitive inhibitor. Therefore, the authors repeated the docking experiment with 2,4-DTBP and proposed a new alternative docking pose with a hydrogen bond. Our new docking indicated a more firm and selective interaction than Sansenya’s report [49]. The new alternative docking pose of 2,4-DTNP provided a better theoretical scenario of anti-glucosidase synergism between DTBP and fatty acids.

Furthermore, in the current study, we performed molecular docking and a re-scoring docking function of all major components based on the GC–MS to evaluate a different energetical pattern between free and esterified fatty acids. Previous reports showed that free fatty acids exhibited more potent inhibitory activity against α-glucosidase than ester forms [30,50,51]. Our finding indicated that free fatty acids have lower energy (more stable) than their esters.

On the other hand, HS-FS-2-M-E showed a unique metabolic profile compared to the other extracts. The major component detected in HS-FS-2-M-E is gluco-octonic acid, and the α-glucosidase inhibition activity of gluco-octonic acid has yet to be discovered. This study is the first report of an in silico gluco-octonic acid’s anti-glucosidase activity. Gluco-octonic acid’s chemical structure is nearly identical to that of glucose. Therefore, the α-glucosidase inhibition of gluco-octonic acid can be expected. However, further in vitro and in vivo experiments must confirm this finding.

The second most abundant metabolite found in HS-FS-2-M-E extract is guanosine. In 2020, Ogasawara et al. [17] reported that guanosine exhibited non-competitive inhibitory behavior against amylase in an in vitro enzyme kinetic assay [17]. Additionally, they reported weak inhibitory activity from guanosine (IC_50_ and Ki values in the mM concentration range). Based on the literature, it is possible that an amylase non-competitive inhibitor can also inhibit α-glucosidase in a similar manner [18]. Therefore, we reported a possible allosteric binding of guanosine on glucosidase and possible molecular interactions between guanosine and amino acids on glucosidase. Our findings on molecular interaction detect several theoretical hydrogen bond interactions, inferring a strong binding. This information contradicted a previous report from Ogasawara et al. [17]. A possible explanation might be that guanosine has a short residence time (1/Koff). Even though guanosine can theoretically firmly interact with α-glucosidase, guanosine might dissociate from α-glucosidase quickly, causing insufficient time to occupy the free enzyme in the system. The speedy dissociation allows the free enzyme to convert substrates to products with little constraint [52]. However, similar to the gluco-octonic acid case, further investigations are required to confirm our prediction on guanosine.

Lastly, the authors detected THC and THCV in the HS-FS-2-M-E fraction. In 2022, Suttithumsatid et al. [53] reported a promising anti-glucosidase activity of THC in a standardized extract. We used their experimental data as a base to predict in silico THCV anti-glucosidase activity using our observed relationship between fatty acid and glucosidase and compared it to THC. We predicted that THCV would likely have weaker activity than THC based on the re-scored total internal energy. Some may argue that the observed relation to fatty acids cannot be used to predict the anti-glucosidase activity of THCV. However, based on the biosynthetic perspective, a distinct chemical structure separating THC and THCV originates from fatty acid biosynthesis [54]. Therefore, the shared biosynthesis between cannabinoids and fatty acids provided an essential ground for applying our observed relationship between fatty acids to THC and THCV. Finally, the molecular simulation suggested that the HS-FS-2-M-E extract may have a different mode of inhibition compared to HS-TH-1-M-E, HS-TH-2-M-E, and HS-FS-1-M-E.

## 5. Conclusions

This study comparatively analyzed the chemical and biological knowledge of *Cannabis sativa* from Thai and foreign samples. More than sixty phytochemicals have been described in hemp seed extracts. Fatty acids exhibited enormous bioactivities. The results showed that all extracts of Thai hemp seed exhibited antibacterial activity, while two extracts of foreign hemp seed were active. The ethanolic extracts of Thai hemp seed (HS-TH-2-M-E) strongly exhibited effective glucosidase inhibitory activity, and their combination with acarbose showed a remarkable synergistic effect on α-glucosidase inhibition. A molecular docking study revealed that fatty acids had the greatest impact on α-glucosidase inhibition. The results from this study provided a strong biochemical rationale for the use of Thai hemp seed as one of the alternative medicines for improving the conditions of diabetic patients.

## Figures and Tables

**Figure 1 foods-13-00055-f001:**
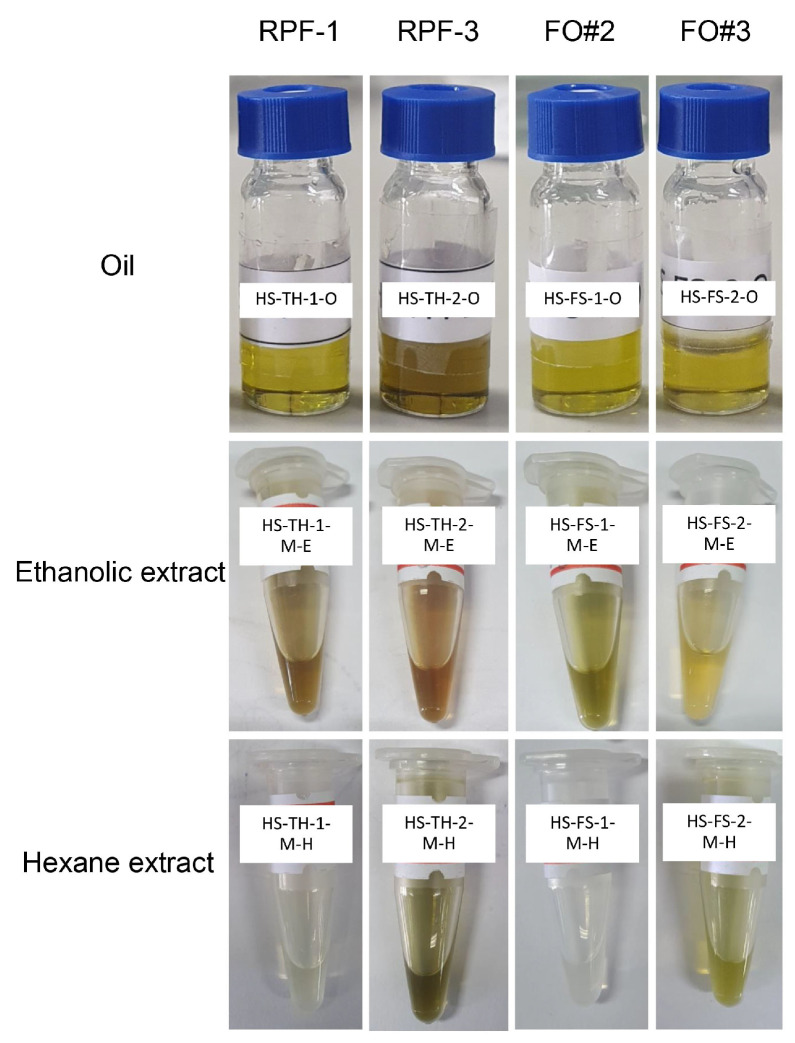
Oils and extracts of hemp (*Cannabis sativa* L.) seed from Thai samples (RPF-1 and RPF-3) and foreign samples (FO#2 and FO#3).

**Figure 2 foods-13-00055-f002:**
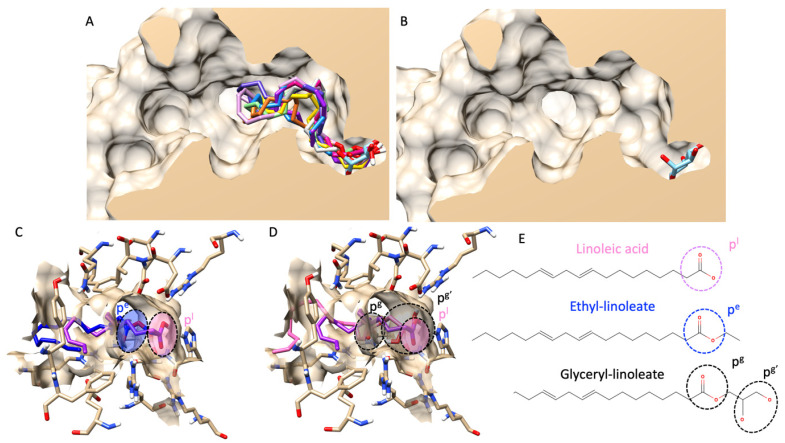
Molecular docking of all fatty acids and their esters. (**A**) shows a structural alignment of the ten most abundant fatty acids and esters in all hemp seed oil extracts (all compounds’ color list is provided in the Appendix A), including a glucose molecule as a native ligand (in a light blue color) from a lateral cross-section of the glucosidase catalytic domain (PDB ID: 3a4a). (**B**) exhibits a similar view to (**A**), but only for a native ligand, glucose. (**C**) demonstrates a 3D visualization of linoleic acid (pink color) and ethyl-linoleate (dark blue color). (**D**) represents an identical view to (**C**) but compares the structural alignment between linoleic acid (pink color) and glyceryl linoleate (black color). (**E**) shows the 2D chemical structures of linoleic acid, ethyl linoleate, and glyceryl linoleate. Circle dash lines indicate the polar head group(s). p^l^ represents a polar head group of linoleic acid. p^e^ represents a polar head group of ethyl linoleate. p^g^ and p^g’^ represent polar head groups of fatty acids and glyceride moieties of glyceryl linoleate.

**Figure 3 foods-13-00055-f003:**
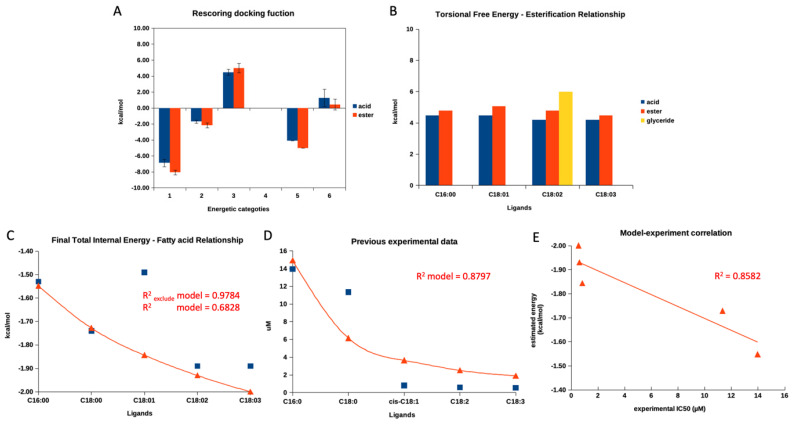
A re-scoring docking function of fatty acids. (**A**) demonstrates a summarized re-scoring docking function in each category, starting from the final intermolecular energy (1), the final total internal energy (2), the torsional free energy (3), the unbound system’s energy (4), and the estimated free energy of binding (5) of five fatty acids and their esters. (**B**) represents the torsional free energy of four fatty acids and their esters. (**C**) shows a relationship between this study’s final total internal energy and five free fatty acids. (**D**) exhibits a relationship between previous experimental IC_50_ values and five free fatty acids. (**E**) represents the current study’s established mathematical model and the previous experimental report. Blue square symbols indicate obtained values from this study, simulation, or previous experiment. Red triangle symbols indicate a calculated value from models, and red lines show a calculated value trend. Only the red triangle symbols in (**E**) represent a coordination of a calculated value obtained from a model and an actual experimental value of each fatty acid.

**Figure 4 foods-13-00055-f004:**
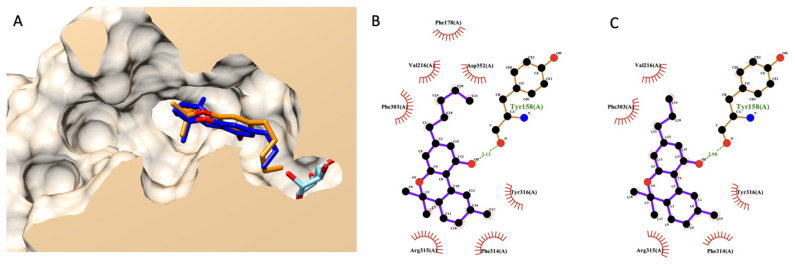
Molecular docking of THC and THCV. (**A**) presents a structural alignment of THC (dark blue), THCV (orange), and a native glucose ligand (light blue color) from a lateral cross-section of the glucosidase catalytic domain (PDB ID: 3a4a). (**B**,**C**) exhibit a 2D interaction diagram of THC and THCV with amino acids in the glucosidase catalytic domain.

**Figure 5 foods-13-00055-f005:**
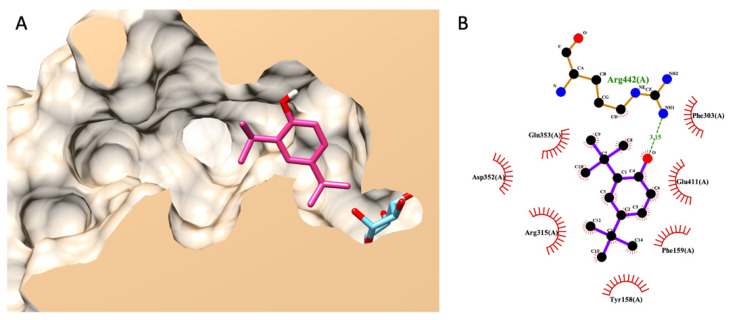
Molecular docking of DTBP. (**A**) exhibits a 3D diagram of DTBP (pink) and native glucose (light blue) ligands from a lateral cross-section of the glucosidase catalytic domain (PDB ID: 3a4a). (**B**) represents a 2D interaction diagram of DTBP and amino acids in the catalytic pocket of glucosidase.

**Figure 6 foods-13-00055-f006:**
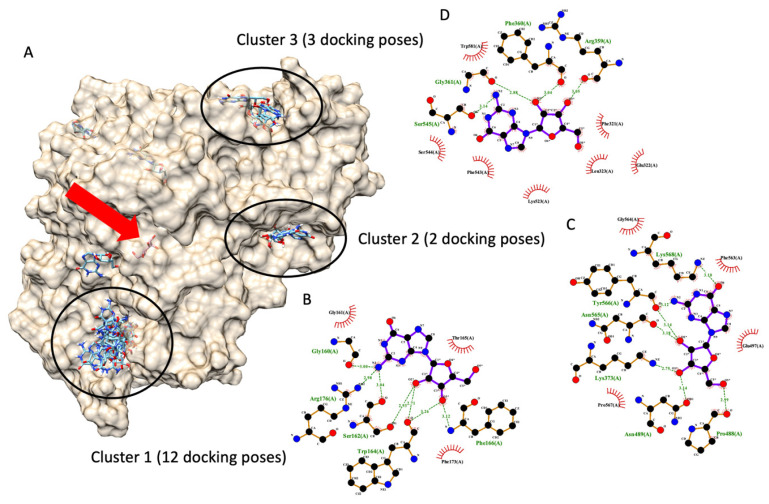
Molecular docking of guanosine. (**A**) demonstrates six docking pockets (clusters) of guanosine on α-glucosidase (PDB ID: 3a4a). The most promising pockets (cluster) with the top three highest docking poses are highlighted via dark circles. The red arrow indicates the native glucose ligand inside the glucosidase catalytic site. (**B**–**D**) exhibited a 2D interaction diagram of a representative docking pose, lowest docking energy, from each cluster ((**B**) for cluster 1, (**C**) for cluster 2, and (**D**) for cluster 3) and amino acid in the glucosidase catalytic domain.

**Figure 7 foods-13-00055-f007:**
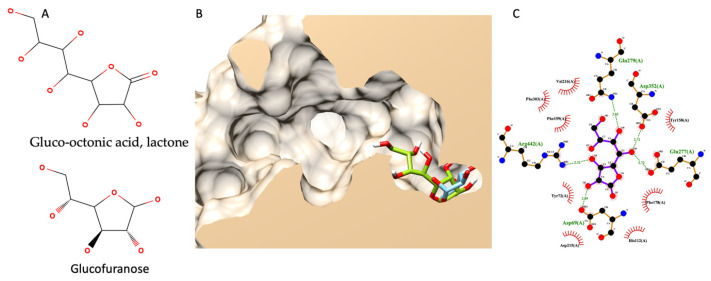
Molecular docking of gluco-octonic acid lactone. (**A**) shows the chemical structure of gluco-octonic acid at the top and glucofuranose at the bottom. (**B**) demonstrates a 3D lateral cross-section of the glucosidase catalytic site (PDB ID: 3a4a) with native glucose (light blue) and selected gluco-octonic acid docking poses (green). (**C**) represents a 2D intermolecular diagram of gluco-octonic acid and amino acid residues in the catalytic domain of glucosidase.

**Table 1 foods-13-00055-t001:** Chemical composition of hemp seed oils and extracts.

Chemicals	% of the Total of Each Chemical Constituent in Each Sample											
	HS-TH-1-O	HS-TH-1-M-H	HS-TH-1-M-E	HS-TH-2-O	HS-TH-2-M-H	HS-TH-2-M-E	HS-FS-1-O	HS-FS-1-M-H	HS-FS-1-M-E	HS-FS-2-O	HS-FS-2-M-H	HS-FS-2-M-E
Palmitic acid	2.05	0.28	9.94	2.74	7.56	4.43	-	10.21	5.58	-	-	1.60
Ethyl palmitoleate	-	-	-	-	-	1.56	-	-	-	-	-	-
Palmitic acid, ethyl ester	-	2.25	4.51	-	1.52	7.14	-	-	8.57	-	1.18	-
α-Linolenic acid (omega-3)	-	-	-	-	-	-	-	-	-	-	-	1.52
Oleic Acid	-	-	-	-	-	-	3.51	-	-	-	-	-
Linoleic acid (omega-6)	20.09 *	22.93 *	35.13 *	86.53 *	66.24 *	34.08 *	-	17.63 *	15.04	-	-	1.43
trans-Oleic acid	16.42	-	-	-	-	-	-	-	5.36	-	-	-
Linoleic acid ethyl ester	-	7.89	10.36	-	-	20.61	-	1.58	11.49	-	1.08	-
Linolenic acid, ethyl ester	-	-	12.27	-	-	-	-	-	-	-	-	-
Ethyl Oleate	-	-	-	-	-	13.02	-	-	10.72	-	-	-
Stearic acid	-	-	-	-	-	-	-	1.73	-	-	-	-
Stearic acid ethyl ester	-	-	3.92	-	-	2.37	-	-	2.65	-	-	-
2-Pentylfuran	-	-	-	-	-	-	-	1.58	-	-	-	-
Glycerin	-	-	-	-	-	1.21	-	-	-	-	-	-
(±)-Glycidol	-	-	-	-	-	-	-	-	-	-	-	1.58
(2R,4R)-2,4 imethyl-1-heptanol	-	-	-	-	-	-	-	1.23	-	-	-	-
4H-Pyran-4-one, 2,3-dihydro-3,5-dihydroxy-6- methyl-	-	-	-	-	-	-	-	-	-	-	-	3.19
2-Isopropyl-5-methyl-6-oxabicyclo[3.1.0]hexane-1-carbaldehyde	-	-	-	-	-	-	-	1.83	-	-	-	-
(2E,4E)-2,4-Decadienal	-	6.94	-	-	-	-	-	8.07	-	-	-	-
5-Pentyl-2(5H)-furanone	-	-	-	-	-	-	-	1.27	-	-	-	-
7-Ethyl-4-nonanone	-	1.95	-	-	-	-	-	3.73	-	-	-	-
Guanosine	-	-	-	-	-	-	-	-	-	-	-	18.58
2,4-Di-tert-butylphenol	-	-	-	-	-	3.19	-	-	-	-	-	-
Benzoic acid, 4-ethoxy-, ethyl ester	-	1.30	-	-	-	-	-	1.21	-	-	1.41	-
Tyramine	-	-	1.45	-	-	-	-	-	-	-	-	-
Crocetane	-	-	-	-	-	-	-	-	-	-	1.03	-
Myo inositol	-	-	-	-	-	-	-	-	-	-	-	5.84
α,β-Gluco-octonic acid lactone	-	-	-	-	-	-	-	-	-	-	-	30.39 *
N-(2-Furylmethyl)-2-methylanilin	-	-	-	-	-	-	-	-	-	-	-	1.31
1-Octadecanol	-	1.39	-	-	-	-	-	1.17	-	-	1.67	-
1-Docosanol	-	-	-	-	-	-	-	1.06	-	-	1.95	-
2-Palmitoylglycerol	-	-	-	-	-	-	-	-	1.08	-	-	-
3-Amino-2-methyl-3-(4-methylphenyl)-1-phenyl-1-propanol	-	-	-	-	-	1.97	-	-	-	-	-	-
Linoleic acid, TMS	-	-	2.24	1.15	-	-	-	-	2.86	-	-	-
2-Monoolein	-	-	-	-	-	-	-	2.53	-	-	-	-
glyceryl-linoleate	-	-	8.03	-	-	-	-	-	15.12 *	-	-	-
β-Monolinolein	-	3.78	-	-	-	2.09	-	-	-	-	2.83	-
Nonanoic acid, 9-(3-hexenylidenecyclopropylidene)-, 2-hydroxy-1-(hydroxymethyl)ethyl ester, (Z,Z,Z)-	-	-	3.84	-	-	-	-	-	6.04	-	-	-
Triterpenoid	2.40	-	-	-	-	-	1.66	-	-	2.18	-	-
Nonacosane	-	-	-	-	-	-	-	-	-	1.77	1.14	-
Vitamin E succinate (calcium)	12.63	-	-	-	-	-	14.16	-	-	15.10	1.09	-
Vitamin E	-	-	-	-	-	-	1.25	-	-	-	-	-
Campesterol	-	1.68	-	-	-	-	-	1.61	1.04	-	4.81	-
(3methyl,24R)-ergost-5-en-3-ol	3.78	-	-	-	-	-	7.61	-	-	5.28	-	-
Stigmasterol	-	-	-	-	-	-	1.41	-	-	1.45	-	-
Clionasterol	15.35	6.66	2.75	1.53	1.44	1.55	29.07 *	5.67	3.80	22.32 *	13.42 *	-
(E)-24-Propylidenecholesterol	-	-	-	-	-	-	-	1.73	-	-	-	-
23(Z)-ethylcholestanol	-	-	-	-	-	-	-	-	-	-	1.05	-
(3b,24Z)-Stigmasta-5,24(28)-dien-3-ol	-	-	-	-	-	-	5.95	-	-	4.30	2.08	-
Lanosterol	1.91	-	-	-	-	-	5.22	-	-	4.29	2.10	-
Cycloartenol	1.85	-	-	-	-	-	3.48	-	-	1.74	2.03	-
γ- Sitostenone	1.69	2.13	-	-	1.35	-	-	1.13	-	-	2.81	-
9-Octadecenoic acid, (2-phenyl-1,3-dioxolan-4- yl) methyl ester, cis-	-	-	-	-	-	-	-	1.50	-	-	-	-
Phytyl linoleate	-	-	-	-	-	-	-	-	-	2.86	-	-
Palmitoleyl oleate	2.61	-	-	-	-	-	-	-	-	-	-	-
Trilinolein	-	-	-	-	-	-	-	-	-	-	9.29	-
∆^9^-Tetrahydrocannabivarin (THCV)	1.20	-	-	-	-	-	-	-	-	5.62	1.05	2.73
2-(6-Isopropenyl-3-methyl-2-cyclohexen-1-yl)-5-pentyl-1,3-benzenediol	-	-	-	-	-	-	-	-	1.04	-	-	-
Cannabidiol (CBD)	0.35	-	-	-	-	-	13.39	-	-	0.53	-	-
∆^9^THC (Dronabino)l	3.36	-	-	-	-	-	-	-	-	18.20	2.70	3.81
Cannabinol (CBN)	1.02	0.51	-	-	-	-	-	-	-	-	-	-

Results are reported as % of the total of each chemical constituent in each sample where an experiment was performed and an analysis of one injection of the test sample was performed. *: major compound.

**Table 2 foods-13-00055-t002:** Minimum inhibitory concentration (MIC) and minimum bactericidal concentration (MBC) values of the active oils and extracts from hemp seed.

Bacteria	MIC/MBC Values (µg/mL)										
	HS-TH-1-O	HS-TH-1-M-E	HS-TH-1-M-H	HS-TH-2-O	HS-TH-2-M-E	HS-TH-2-M-H	HS-FS-1-M-E	HS-FS-2- M-E	Oxacillin	Vancomycin	Norfloxacin
SA	2048/>2048	512/>2048	2048/>2048	128/>2048	512/>2048	256/>2048	-	-	0.125/0.50	ND	ND
SE	-	128/>2048	-	256/>2048	256/>2048	-	256/>2048	2048/>2048	0.125/0.50	ND	ND
MRSA	-	512/>2048	-	256/>2048	1024/>2048	512/>2048	-	-	ND	0.5/1	ND
CA	-	512/512	-	256/512	512/512	256/256	2048/2048	1024/1024	0.25/0.25	ND	ND
EC	-	-	-	-	-	-	-	-	ND	ND	2/4
PA	-	-	-	-	-	-	-	-	ND	ND	4/8

SA: *Staphylococcus aureus* (ATCC 25923), SE: *Staphylococcus epidermidis* (TISTR 517), MRSA: Methicillin-resistant *Staphylococcus aureus* (DMST 20654), EC: *Escherichia coli* (ATCC 25922), PA: *Pseudomonas aeruginosa* (ATCC 27853), CA: *Cutibacterium acnes*, HS: Hemp seed; TH: Thailand species RPF-1 (HS-TH-1) and RPF-3 (HS-TH-2); FS: Foreign species FO#2 (HS-FS-1) and FO#3 (HS-FS-2); O: Hemp seed oil; M: Marc, H: Hexane, E: 80% Ethanol, ND: not detected.

**Table 3 foods-13-00055-t003:** α-glucosidase inhibition of oils and extracts from hemp seed at 2 and 0.25 mg/mL and IC_50_.

No.	Sample	Extract Part	Code	% Inhibition ± SD		IC_50_ (µg/mL)
				2 mg/mL	0.25 mg/mL	
1.	RPF-1	Oil	HS-TH-1-O	15.13 ± 2.85		-
		Marc	HS-TH-1-M-H	18.46 ± 2.75		-
			HS-TH-1-M-E	98.51 ± 0.14	87.34 ± 1.23 *	42.72
2.	RPF-3	Oil	HS-TH-2-O	16.94 ± 3.02		-
		Marc	HS-TH-2-M-H	15.97 ± 3.45		-
			HS-TH-2-M-E	97.99 ± 0.31	85.58 ± 0.07 *	33.27
3.	FO#2	Oil	HS-FS-1-O	5.66 ± 1.28		-
		Marc	HS-FS-1-M-H	20.68 ± 4.70		-
			HS-FS-1-M-E	100.36 ± 2.4	75.93 ± 2.40 *	94.09
4.	FO#3	Oil	HS-FS-2-O	14.78 ± 0.46		-
		Marc	HS-FS-2-M-H	10.95 ± 2.63		-
			HS-FS-2-M-E	98.99 ± 0.08	83.83 ± 1.10 *	67.44
Control	Acarbose			93.95 ± 1.07	47.40 ± 0.66	285.92

Results are reported as means of % inclusion ± SD where the experiment was performed in triplicate (* *p* < 0.05 compared to acarbose).

**Table 4 foods-13-00055-t004:** α-glucosidase inhibition and CI for combinations of acarbose with the HS-TH-2-M-E.

Acarbose IC_50_ Value Ratio	HS-TH-2-M-E IC_50_ Value Ratio				
	0.25	0.50	1	2	4
4	0.90	0.70	0.71	0.80	0.96
2	0.96	0.95	0.99	0.96	0.99
1	2.77	1.30	1.06	0.92	0.85
0.50	1.39	1.36	1.21	0.94	0.93
0.25	1.83	1.01	0.97	0.81	0.83

**Table 5 foods-13-00055-t005:** Detailed re-scoring docking functions of selected THC and THCV poses.

No	Energy	THC	THCV
1	Final intermolecular Energy (kcal/mol)	−5.42	−5.48
	vdW + Hbond + desvol Energy	−5.33	−5.29
	Electrostatic Energy	−0.09	−0.2
2	Final Total Internal Energy (kcal/mol)	−0.41	−0.20
3	Torsional Free Energy (kcal/mol)	+1.49	+0.89
4	Unbound System’s Energy (kcal/mol)	0.00	0.00
5	Estimate the Free Energy of Binding (kcal/mol)	−4.34	−4.72
6	Estimate Ki (mM)	0.66	0.34

## Data Availability

Data is contained within the article or Appendix A.

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
