# Peer review of "A Comparative Study of Chemical Profiling and Bioactivities between Thai and Foreign Hemp Seed Species (Cannabis sativa L.) Plus an In-Silico Investigation"

_foods, 2023, doi:10.3390/foods13010055_

Round 1

Reviewer 1 Report

Comments and Suggestions for Authors

The MS is well prepared and is hypothesis driven. 

How many compounds were evaluated in silico?

Why these enzymes target were selected?

Provide statistical analysis of data generated compared with control

Author Response

Reply to Reviewer 1

Comments and Suggestions for Authors

The MS is well prepared and is hypothesis driven.

### Thank you so much.

How many compounds were evaluated in silico?

Response: In silico study, the authors evaluated of 15 compounds in molecular docking, there were five free fatty acids (palmitic acid, stearic acid, oleic acid, linoleic acid, and linolenic acid), five fatty acid esters (ethyl-palmitate, ethyl-oleate, ethyl linoleate, glyceryl-linoleate, and ethyl-linoleate), two cannabinoids (THC and THCV), one phenolic compound (DTBP), and two glycosides (guanosine and gluco-octonic acid lactone).

Why these enzymes target were selected?

Response: Glucosidase is found on the brush-border surface membrane of intestinal cells. It increases postprandial blood glucose levels by catalyzing the hydrolysis of oligosaccharides and disaccharides into monosaccharides. Thus, α-glucosidase is a desirable target for the treatment of type-2 diabetes mellitus. Acarbose is the first marketed α-glucosidase inhibitor. However, it can lead to side-effects. For this reason, it is also essential to identify α-glucosidase inhibitors with high selectivity, good activity, low toxicity, and low cost.

Provide statistical analysis of data generated compared with control

Response: Revised accordingly in Table 3. The stat analysis was used to compare the % inhibition of the active extracts with the positive control (acarbose)

### Thank you for all suggestions, we tried to correct and improve our manuscript and update it already.

Best Regards,

Authors

Reviewer 2 Report

Comments and Suggestions for Authors

Sangkanu and colleagues prepared a manuscript on comparative study of chemical profiling and bioactivities between Thai and Foreign Hemp Seed Species. The title of the article is quite controversial (plus In Silico Investigation), but I leave this to the discretion of the editor and authors. As stated in the abstract and introduction, their focus is characterizing the chemical profile of 12 hemp seed extracts from Thai and international samples using gas chromatography-mass spectrometry and investigate their antibacterial properties and α-glucosidase inhibition. Material and method section is sufficient to get an insight into methodology used and is correctly described. Results are supported by the obtained data. Discussion is appropriate. I recommended that this manuscript minor revision. Further efforts are required for improving the quality, grammar and coherence of the manuscript. There are some minor typos in the text, for example alpha or α-, it must be uniform, unnecessary hyphens and periods, etc.

Author Response

Reply to Reviewer 2

Comments and Suggestions for Authors

Sangkanu and colleagues prepared a manuscript on comparative study of chemical profiling and bioactivities between Thai and Foreign Hemp Seed Species. The title of the article is quite controversial (plus In Silico Investigation), but I leave this to the discretion of the editor and authors. As stated in the abstract and introduction, their focus is characterizing the chemical profile of 12 hemp seed extracts from Thai and international samples using gas chromatography-mass spectrometry and investigate their antibacterial properties and α-glucosidase inhibition. Material and method section is sufficient to get an insight into methodology used and is correctly described. Results are supported by the obtained data. Discussion is appropriate. I recommended that this manuscript minor revision. Further efforts are required for improving the quality, grammar and coherence of the manuscript. There are some minor typos in the text, for example alpha or α-, it must be uniform, unnecessary hyphens and periods, etc.

Response: Revised alpha to α.

### Thank you for all suggestions, we tried to correct and improve our manuscript and update it already.

Best Regards,

Authors

Reviewer 3 Report

Comments and Suggestions for Authors

1. You discuss differences in the colors of your hemp seed extracts are you able to explain why there are differences? 

2. There is no need to have RT (retention time) in the table unless you actually list the RT. The data shown in the VOC table is this an average of multiple runs or just one. The legend at the bottom needs to explain what this data represents. Also did you have standard deviation for this data. How many samples were run. Was the data found what you expected? You have this huge table briefly mention it. 

3. My assumption is in Table 3 that the data is a Average+/-STD. Make sure to say that in the legend of the table. Tables should be able to stand on their own without having to read a ton of text. 

4. Why did you not run any simple stats on your data? Why based upon the data I can see there are differences it would be nice to have some data 

Comments on the Quality of English Language

The English is rather clunky in areas and needs work to help improve the overall flow of the manuscript. 

Author Response

Reply to Reviewer 3

Comments and Suggestions for Authors

  1. You discuss differences in the colors of your hemp seed extracts are you able to explain why there are differences?

Response: Revised in discussion. “The green color of hemp seed oil extracts comes from the most important pigments, especially chlorophyll a and b. Furthermore, carotenoids are also presented in the hemp seed oil. However, the ratio of the carotenoid/chlorophyll proportion varied significantly and depending on the HSO seed variety.”

Reference: [33] Blasi, F., Tringaniello, C., Verducci, G., & Cossignani, L. (2022). Bioactive minor components of Italian and Extra-European hemp seed oils. Lwt, 158, 113167.

  1. There is no need to have RT (retention time) in the table unless you actually list the RT. The data shown in the VOC table is this an average of multiple runs or just one. The legend at the bottom needs to explain what this data represents. Also did you have standard deviation for this data. How many samples were run. Was the data found what you expected? You have this huge table briefly mention it.

Response: Revised accordingly in remark of Table 1. “Results are reported as % of total of each chemical constituent in each sample where experiment was performed analysis of one injection of the test samples”.

  1. My assumption is in Table 3 that the data is a Average+/-STD. Make sure to say that in the legend of the table. Tables should be able to stand on their own without having to read a ton of text.

Response: Revised accordingly in remark of Table 3 “Results are reported as means of               % inhibition ± SD where experiment was performed in triplicate (*p < 0.05 compared to acarbose)”.

  1. Why did you not run any simple stats on your data? Why based upon the data I can see there are differences it would be nice to have some data.

Response: Revised accordingly in Table 3. The stat analysis was used to compare the % inhibition of the active extracts with the positive control (acarbose).

Comments on the Quality of English Language

The English is rather clunky in areas and needs work to help improve the overall flow of the manuscript.

### Thank you for all suggestions, we tried to correct and improve our manuscript and update it already.

Best Regards,

Authors

Reviewer 4 Report

Comments and Suggestions for Authors

Dej-adisai et al. reported a comprehensive study on the chemical profile and bioactivities of 12 Thai and Foreign Hemp Seed Species (Cannabis sativa L.), a plant widely used by humans for textiles, food, and medicine. To accomplish these purposes, the authors applied gas chromatography-mass spectrometry (GC-MS), combined with the evaluation of their antibacterial activity, α-glucosidase inhibitory activity an in silico protocol. Linoleic acid was a major component presented in Thai hemp seed extracts, while α,β-gluco-octonic acid lactone, clionasterol and glyceryl-linoleate were detected as the main metabolites found in foreign hemp seed extracts. Furthermore, eight extracts from both Thai and foreign hemp seeds exhibited antibacterial activity against Staphylococcus aureus, Staphylococcus epidermidis, Methicillin-resistant Staphylococcus aureus, and Cutibacterium acnes. Interestingly, the ethanol extract of Thai hemp seed (HS-TH-2-M-E) showed a superior α-glucosidase inhibition over foreign species. The combination of Thai hemp species (HS-TH-2-M-E) and acarbose showed a synergistic effect against α-glucosidase. Furthermore, the docking investigation revealed that fatty acids had a greater impact on α-glucosidase than fatty acid esters and cannabinoids. The computational simulation predicts a potential allosteric binding pocket of guanosine on glucosidase and is the first description of gluco-octonic acid's anti-glucosidase activity in silico.

The findings concluded that Thai hemp seed could be used as a resource for supplemental drug or dietary therapy for diabetes mellitus.

On these premises, even if this manuscript could be of interest to the readership of Foods, I believe that the manuscript to meet the journal standard, some major revisions are needed.

In my opinion, the main issues of this manuscript are related to the in silico part.

Specifically, even if there is a detailed description of the computational results, some parts should be improved and/or simplified or deleted. In more detail, it is not very clear the reason related to the introduction of the mathematical equation instead of an improvement of experimental procedures; specifically, as reported by Rupesh Agarwal and Jeremy C. Smith (doi.org/10.1002/minf.202200188), a correct balance between “Speed vs Accuracy: Effect on Ligand Pose Accuracy of Varying Box Size and Exhaustiveness in AutoDock Vina ” could improve the reliability of the results. Furthermore, the redocking scoring function is not described in detail from a mathematical point of view, so it is not clear which is the correction applied.

However, in my opinion, the authors should increase exhaustiveness and different grid box sizes, and after a detailed comparison and/or analysis, they should justify and introduce a “mathematical manipulation” of the results.

Moreover, as reported in detail by Bedi et al. in 2023 (https://doi.org/10.1016/j.molstruc.2023.135115) “Recent developments in synthetic α-glucosidase inhibitors: A comprehensive review with structural and molecular insight”, where it is evident the influence of hydrophobic, hydrogen and other types of non-bonded interactions, and so it could be possible to rationalize and/or predict the difference of activities applying a common combination between a qualitative and quantitative type of analysis.

Moreover, to suggest and/or identify novel pockets, the authors should use AutoLigand “a tool to identify ligand binding sites on or within receptor proteins. AutoLigand uses an effective method to scan rapidly for high affinity binding pockets and reports the optimal volume, shape, and best atom types for the identified ligand binding sites” (https://autodock.scripps.edu/resources/autoligand/)

In this present form, it is difficult for readers to understand the rationale behind the identification of these novel potential binding sites that also could not be corroborated by experimental investigation.

Minor issues:

 1-     In Table 1 and line 271 the authors report the identification of Dronabinol, but this is a synthetic compound. Therefore, since it is the GC-MS analysis of a natural extract and EO, it is necessary to use the term D9THC.

2-     The doi could be reported for each reference

 In conclusion, the current version of the manuscript does not fully satisfy all the aspects and the high impact standards required by the aim of the work and by the Foods, and I recommend major revisions.

Comments on the Quality of English Language

Minor editing of English language required

Author Response

Reply to Reviewer 4

Comments and Suggestions for Authors

Dej-adisai et al. reported a comprehensive study on the chemical profile and bioactivities of 12 Thai and Foreign Hemp Seed Species (Cannabis sativa L.), a plant widely used by humans for textiles, food, and medicine. To accomplish these purposes, the authors applied gas chromatography-mass spectrometry (GC-MS), combined with the evaluation of their antibacterial activity, α-glucosidase inhibitory activity an in silico protocol. Linoleic acid was a major component presented in Thai hemp seed extracts, while α,β-gluco-octonic acid lactone, clionasterol and glyceryl-linoleate were detected as the main metabolites found in foreign hemp seed extracts. Furthermore, eight extracts from both Thai and foreign hemp seeds exhibited antibacterial activity against Staphylococcus aureus, Staphylococcus epidermidis, Methicillin-resistant Staphylococcus aureus, and Cutibacterium acnes. Interestingly, the ethanol extract of Thai hemp seed (HS-TH-2-M-E) showed a superior α-glucosidase inhibition over foreign species. The combination of Thai hemp species (HS-TH-2-M-E) and acarbose showed a synergistic effect against α-glucosidase. Furthermore, the docking investigation revealed that fatty acids had a greater impact on α-glucosidase than fatty acid esters and cannabinoids. The computational simulation predicts a potential allosteric binding pocket of guanosine on glucosidase and is the first description of gluco-octonic acid's anti-glucosidase activity in silico.

The findings concluded that Thai hemp seed could be used as a resource for supplemental drug or dietary therapy for diabetes mellitus.

On these premises, even if this manuscript could be of interest to the readership of Foods, I believe that the manuscript to meet the journal standard, some major revisions are needed.

In my opinion, the main issues of this manuscript are related to the in silico part.

Specifically, even if there is a detailed description of the computational results, some parts should be improved and/or simplified or deleted. In more detail, it is not very clear the reason related to the introduction of the mathematical equation instead of an improvement of experimental procedures; specifically, as reported by Rupesh Agarwal and Jeremy C. Smith (doi.org/10.1002/minf.202200188), a correct balance between “Speed vs Accuracy: Effect on Ligand Pose Accuracy of Varying Box Size and Exhaustiveness in AutoDock Vina ” could improve the reliability of the results. Furthermore, the redocking scoring function is not described in detail from a mathematical point of view, so it is not clear which is the correction applied.

However, in my opinion, the authors should increase exhaustiveness and different grid box sizes, and after a detailed comparison and/or analysis, they should justify and introduce a “mathematical manipulation” of the results.

Moreover, as reported in detail by Bedi et al. in 2023 (https://doi.org/10.1016/j.molstruc.2023.135115) “Recent developments in synthetic α-glucosidase inhibitors: A comprehensive review with structural and molecular insight”, where it is evident the influence of hydrophobic, hydrogen and other types of non-bonded interactions, and so it could be possible to rationalize and/or predict the difference of activities applying a common combination between a qualitative and quantitative type of analysis.

To address the reviewer’ comment:

First of all, the authors would like to thank the reviewer for your suggestion. However, based on the suggested article by the reviewer, the paper showed that increasing exhaustiveness and reducing the grid box can improve the pose by looking at the RMSD value. However, looking at the result, increasing exhaustiveness can only slightly improve RMSD (0.6 to 0.1 Å) in most cases (comparing between 8, 25, 50, 75, and 100). Generally, it is known that increasing exhaustiveness will cause more computational power and time. Therefore, the authors of the suggested article by the reviewer recommended the exhaustiveness of 8 as a default parameter because it provided an accepted outcome with reasonable computational resources.

In the authors of this manuscript's defense, we used an exhaustiveness of 10, higher than a recommendation.

For the grid size parameter, it is also known that reducing this parameter will improve the RMSD value. However, it also comes with a significant limitation. Reducing grid size will reduce docking (searching) space, not allowing ligands to move flexibly. Therefore, it could lead to a false docking pose, especially when no experimental structure exists to compare, like compounds of the authors’ interest (fatty acids and their derivatives). In general, as well as to the authors' best knowledge, there is no standard value of the grid size, but using values that are too small and too large will raise questions on the docking result. Therefore, as a standard protocol, it is essential to validate the establishment of a docking protocol using a native ligand (comes with crystal protein) since it can be used to validate the established docking protocol. In the manuscript, the authors validated the docking protocol, and the validation passed the acceptance criteria (re-dock RMSD less than 3 Å, DOI: 10.3390/ph14090896).

In conclusion, to answer the reviewer’s question, repeating the docking experiment by increasing 2.5x time exhaustiveness to 25 (causing more computational and waiting time) and reducing grid size (limiting the search space, rigid docking) is likely not the solution for the review concern based on the input resources and expected outcome from the reviewer’ suggested.

In fact, the authors are currently conducting a computational experiment using molecular dynamic techniques to solve this problem. As mentioned in the paper of Hugo Guterres and Wonpil Im (DOI: doi.org/10.1002/pro.4413), a short MD simulation could significantly improve the predicted pose correctness from 0.6 to 0.8. It is also known that MD simulation is much more reliable than docking. However, the authors cannot wait for the MD experiment since our manuscript is condensed. Therefore, the authors decided to publish the manuscript in the current version.

To address the reviewer's concern about the mathematical model, first, the authors would like to inform the reviewer that we aim to use re-docking energetical scores (or predicted binding energy) to examine the correlation between computational and experimental outcomes. This could not be achieved by using RMSD values. The reason that the authors had to perform a re-scoring experiment was because Vina's predicted binding energy was not correlated with experimental data. The same finding was also reported by NT Nguyen et al. (2019) (DOI: 10.1021/acs.jcim.9b00778). Additionally, NT Nguyen and team discovered that autodock4 (AD4) provided a higher correlation between a predicted binding energy and experimental data (correlation coefficient 0.56 for AD4 and 0.49 for vina). Therefore, the authors used AD4 to re-calculate predicted binding energy. However, even AD4 did not provide a satisfactory correlation in our case. Therefore, the authors took a further step to analyze detailed energies contributing to AD4-predicted binding energy. As mentioned in the manuscript, the authors only found that calculated final total internal energy was better correlated to experimental data than AD4 re-calculated predicted binding energy (Table S5 in supplementary file and Table a1 below).

Table a1 Correlation values between experimental data and re-scoring function obtained from AD4.

No.

Name

Literature

1

2

3

4

5

IC50 (uM)

Final Intermolecular Energy (kcal/mol)

Final Total Internal Energy (kcal/mol)

Torsional Free Energy (kcal/mol)

Unbound System's Energy (kcal/mol)

Estimate Free Energy of Binding (kcal/mol)

1

C16:0

13.97

-7.59

-1.53

4.47

0

-4.65

2

C18:0

11.34

-7.11

-1.74

5.07

0

-3.77

3

cis-C18:1

0.81

-6.43

-1.49

4.47

0

-3.45

4

C18:2

0.6

-6.65

-1.89

4.18

0

-4.36

5

C18:3

0.54

-6.65

-1.89

4.18

0

-4.36

R2

1.0000

-0.9500

0.4117

0.6593

N.D.

-0.2465

Generally, to further confirm this docking finding, MD simulation through MMPBSA is required, as suggested by Hugo Guterres and Wonpil Im (DOI: doi.org/10.1002/pro.4413). Again, for the same reason as mentioned above, the authors cannot wait for the MD simulation (currently ongoing experiment) to add data to the manuscript since the manuscript is already dense. However, the authors could not leave the point unsolved. Therefore, the authors used the mathematical model to correct the calculated energy by a docking program. A similar concept has been used widely in the field, as reported by K Crampon et al. (2022), that machine learning, including linear regression mathematical model, has been used to improve docking predictive binding energy (AD da Silva et al. (2022), DOI: 10.1002/jcc.26048). Even though the authors did not create a new regression model to correct the predictive binding score, we used a non-linear regression to analyze the correlation qualitatively and quantitatively between calculated final total internal energy and experimental data. As shown in the manuscript, the authors found that both calculated final total internal energy by docking program and experimental data exhibited a virtually exponential decay function. To confirm this, the authors used online statistical software to fit the obtained data with the existing equation provided by the software to confirm the observed virtual pattern (virtually exponential decay function). As a result, the statistic model confirmed that both calculated final total internal energy by docking program and experimental data express an exponential decay function based on an R2 value of 0.68 and 0.85. However, the model obtained from the calculated final total internal energy did not show a satisfactory R2 value because of the final total internal energy of C18:1. Therefore, removing the value of C18:1 improves the correlation value up to 0.97 (Figure 3C in the manuscript). This procedure is common in statistical analysis. Below is an additional table and figure for the reviewer. As presented, using original data (excluding C18:1), the correlation value (R2 of 0.8661, Figure a1 below) is similar to a proposed model in the manuscript (R2 of 0.8582, Figure 3E). Therefore, the authors used the proposed mathematical model (an exponential decay function) to compare it with experimental data instances of original data.

Table a2. Data used of compounds of interest excluded C18:1

Carbon range

Name

Final Total Internal Energy (kcal/mol)

IC50 (uM)

1

C16:00

PA

-1.53

13.97

2

C18:00

SA

-1.74

11.34

4

C18:02

LA

-1.89

0.6

5

C18:03

LNA

-1.89

0.54

Figure a1 Linear relationship between docking and experimental data excluded C18:1

Finally, regarding the reviewer’s suggestion of repeating the docking experiment by adjusting the exhaustiveness and grid size parameters, no information indicates that changing these two parameters will improve predictive binding energy from the docking program, which can correlate better to experimental data based on the recommended article by the reviewers. Therefore, the authors kindly disagree with the reviewer on the repeat docking experiment since our docking protocols are at the standard level used by other international researchers in the field, as provided in the examples above.

However, as also mentioned earlier, the authors are ongoing investigating MD simulation of these fatty acid derivatives, and based on Hugo Guterres and Wonpil Im (DOI: doi.org/10.1002/pro.4413), MD simulation is not only improving ligand pose but also binding energy of ligand-protein binding. Therefore, the authors are convinced that ongoing MD investigation will provide a better outcome than repeating the docking experiment. Again, the authors will present the findings of our MD simulation in our next manuscript, not in the current manuscript.

Regarding the reviewer’s suggested review article, after reading through the article timely, there is no significant information on fatty acids and their derivative list in the publication. More importantly, nearly all compounds listed in the suggested review article are synthetic compounds (except steroids), unlike the authors’ compounds of interest, which are natural products. Therefore, the authors cannot find the link between the suggested review article and our work. Furthermore, the review article did not discuss how the calculated energies like torsion and total internal energies may contribute to glucosidase activity, as the authors discovered in our manuscript. In conclusion, the review article suggested by the reviewer did not correlate with the authors’ manuscript chemically and energetically. Therefore, the authors cannot cite the suggested review article in our manuscript since there is not link to our work.

Moreover, to suggest and/or identify novel pockets, the authors should use AutoLigand “a tool to identify ligand binding sites on or within receptor proteins. AutoLigand uses an effective method to scan rapidly for high affinity binding pockets and reports the optimal volume, shape, and best atom types for the identified ligand binding sites” (https://autodock.scripps.edu/resources/autoligand/)

In this present form, it is difficult for readers to understand the rationale behind the identification of these novel potential binding sites that also could not be corroborated by experimental investigation.

To answer the reviewer's question regarding identifying an allosteric site for guanosine, first, the author introduces the background information and the author's hypothesis of proposing a possible allosteric site for guanosine. A similar paragraph shown below was added to the manuscript.

“In 2020, Ogasawara et al. reported that guanosine exhibited a non-competitive inhibitory behavior against amylase from an in vitro enzyme kinetic assay [17]. This information from Ogasawara and the team indicated that guanosine bound with an enzyme at a non-catalytic pocket, an allosteric site.  From the literature, it is possible that an amylase non-competitive inhibitor can also inhibit glucosidase [18]. Therefore, it is likely that guanosine might inhibit the glucosidase similarly, and this is the authors’ hypothesis. To prove the hypothesis, the authors modified the authors’ docking protocol to cover the entire glucosidase enzyme and perform molecular docking. Since the docking protocol was modified, the authors re-validated it. This particular docking aims to identify the binding pocket of the ligand of interest, not a ligand conformation like a conventional docking. Therefore, the evaluation criterion shifted from an RMDS value of less than 3 Å to the highest docking poses in the same pocket. The modified docking protocol could re-dock a glucose molecule (a native ligand) back into an active site (original pocket) with the highest docking poses. It indicated that the entire enzyme structure docking could be used to predict a ligand binding site. The authors provided information regarding the entire enzyme structure docking validation in a supplementary file.”

Regarding identifying binding using the AutoLigand program, similar to other predicted binding site programs, AutoLigand predicts possible binding sites (usually more than one site is predicted based on the authors' experience) based on the protein structure alone without considering the ligand molecule (R Harris et al. (2009), DOI: 10.1002/prot.21645.). Therefore, based on the principle of the searching approach, predicted outcomes from AutoLigand would not be the answer specifically for guanosine. Therefore, the authors did not use the AutoLigand program.

For instance, the authors adapted a classic docking protocol to search for a possible allosteric site of guanosine on the entire glucosidase enzyme structure because a principle of search relied on identifying the top ligand confirmations with the lowest minima in terms of binding energy on the glucosidase structure. It could refer to a possible binding site of guanosine on glucosidase. This concept was proposed by Autodock developing scientists in 2010 (S Cosconati et al. (2010), DOI: 10.1517/17460441.2010.484460), and it is also known as blind docking. The authors also provided an example of another article using blind docking via the Autodock Vina program published in a respected journal to support our statement in the following statement. Here is an example: a publication from G Grasso and team (DOI: 10.1080/07391102.2021.1988709). Therefore, using blind docking via Autodock Vina to identify a possible binding site of compounds is academically accepted and can be published in a respected journal internationally.

Finally, the authors hope that our answer addresses all major points raised by the reviewer and can convince the reviewer that the authors’ docking experiments are up to the research standard used by international researchers.

Minor issues:

  1. In Table 1 and line 271 the authors report the identification of Dronabinol, but this is a synthetic compound. Therefore, since it is the GC-MS analysis of a natural extract and EO, it is necessary to use the term D9THC.

Response: Thank you so much. The revision has done.

  1. The doi could be reported for each reference

Response: Thank you so much. The revision has done.

In conclusion, the current version of the manuscript does not fully satisfy all the aspects and the high impact standards required by the aim of the work and by the Foods, and I recommend major revisions.

### Thank you for all suggestions, we tried to correct and improve our manuscript and update it already.

Comments on the Quality of English Language

Minor editing of English language required

### Thank you so much, we asked the English specialist to correct our manuscript.

Best Regards,

Authors

Round 2

Reviewer 4 Report

Comments and Suggestions for Authors

The manuscript can be accepted in present form